# Satellite Glial Cells and Neurons in Trigeminal Ganglia Are Altered in an Itch Model in Mice

**DOI:** 10.3390/cells11050886

**Published:** 2022-03-04

**Authors:** Meytal Cohen, Rachel Feldman-Goriachnik, Menachem Hanani

**Affiliations:** 1Laboratory of Experimental Surgery, Hadassah-Hebrew University Medical Center, Jerusalem 91240, Israel; meytal.cohen@mail.huji.ac.il (M.C.); rahel.gor@gmail.com (R.F.-G.); 2Faculty of Medicine, The Hebrew University of Jerusalem, Mount Scopus, Jerusalem 91240, Israel

**Keywords:** sensory ganglion, glial fibrillary acidic protein, gap junction, calcium imaging, purinergic receptor

## Abstract

Itch (pruritus) is a common chronic condition with a lifetime prevalence of over 20%. The mechanisms underlying itch are poorly understood, and its therapy is difficult. There is recent evidence that following nerve injury or inflammation, intercellular communications in sensory ganglia are augmented, which may lead to abnormal neuronal activity, and hence to pain, but there is no information whether such changes take place in an itch model. We studied changes in neurons and satellite glial cells (SGCs) in trigeminal ganglia in an itch model in mice using repeated applications of 2,4,6-trinitro-1-chlorobenzene (TNCB) to the external ear over a period of 11 days. Treated mice showed augmented scratching behavior as compared with controls during the application period and for several days afterwards. Immunostaining for the activation marker glial fibrillary acidic protein in SGCs was greater by about 35% after TNCB application, and gap junction-mediated coupling between neurons increased from about 2% to 13%. The injection of gap junction blockers reduced scratching behavior, suggesting that gap junctions contribute to itch. Calcium imaging studies showed increased responses of SGCs to the pain (and presumed itch) mediator ATP. We conclude that changes in both neurons and SGCs in sensory ganglia may play a role in itch.

## 1. Introduction

Itch (pruritus) has been defined as “an unpleasant cutaneous sensation that provokes the desire to scratch” [1]. This is a very common chronic condition affecting over 20% of the population during their lifetime [2], and it is difficult to treat [3,4,5]. Pruritus is associated with many skin diseases, the most prevalent of which is atopic dermatitis [6]. Pruritus is frequently present in systemic disease (renal, hepatic, and others) and is also caused by numerous medications [1]. Pruritus has a major detrimental influence on the quality of life, which is comparable to that of chronic pain, requiring the use of tranquilizers or antidepressants [1,7]. Chronic itch may lead to repetitive scratching that causes skin lesions, which can worsen the itch sensation and lead to further scratching (the itch–scratch cycle).

Histamine has long been recognized as a common pruritogen and it activates a subgroup of sensory neurons via H1 receptors [3,5]. A major progress in itch research took place with the identification of a family of receptors called Mrgprs, which mediate responses to a number of non-histaminergic pruritogens [8,9,10]. Itch receptors (H1, Mrgprs, and other non-H1 receptors) are all found in a variety of pain sensitive neurons. It has been reported that pruriceptors are a subgroup of nociceptors and there is some overlap between pain and itch pathways [1,3,10,11,12]. It has been found that many pain-sensing sensory neurons can also transmit itch [11]. Pain and itch are distinct sensations, although itch is usually accompanied by a weak sensation of pain [13]. A number of theories have been advanced to explain itch signaling, but this field remains controversial [10,11,12].

Although there is an emphasis on central processing in pain and itch research, it is widely accepted that peripheral mechanisms also play a role in these sensations. There is considerable evidence that sensory ganglia are important in the generation and maintenance of chronic pain [14,15]. In contrast, there is little information in the literature on the role of sensory ganglia in chronic itch.

Research over the past 20 years has provided firm support for the contribution of glial cells (astrocytes and microglia) in the spinal cord to chronic pain [16,17,18,19]. There is emerging evidence that the activation of spinal astrocytes (astrogliosis) and microglia also plays a role in itch [1,16,17,18,19]. The main type of glial cells in sensory ganglia are satellite glial cells (SGCs), which form a complete envelope around neuronal somata. These cells were found to make important contributions to the abnormal electrical activity of sensory neurons and, therefore, to nociception [20,21,22], but there is no information on their possible role in itch.

Kitagaki et al. [23] described a model for atopic dermatitis based on repeated application of 2,4,6-trinitro-1-chlorobenzene (TNCB) in the ears of mice, which caused skin inflammation. They found that this treatment resulted in strong scratching behavior. The model shared many of the histopathological, immunological, and clinical features of human atopic dermatitis. To explore the possible role of SGCs in pruritus, we investigated, in this model, how neurons and SGCs in the mouse trigeminal ganglion (TG) were altered. The results indicate that these cells undergo several changes in this model that are similar in some (but not all) aspects to those that take place in these cells in pain models.

## 2. Materials and Methods

### 2.1. Subjects

For this study, we used Balb/c mice, 2–5 months old, of either sex (males/females about 1:1, unless otherwise indicated), weighing 20–25 g. The animals were housed under controlled light (07:00–19:00) and temperature (21–24 °C) conditions and provided with food and water ad libitum. All procedures were approved by the Animal Care and Use Committee of the Hebrew University Hadassah Medical School, and conformed to the National Institutes of Health standards for the care and use of laboratory animals.

### 2.2. Tracing Trigeminal Ganglia (TG) Neurons

To trace the projection of TG neurons to the ear, we injected the ears on both sides with fluorescent tracer 1,1′-dioctadecyl- 3,3,3′,3′-tetramethylindocarbocyanine perchlorate (DiI, Molecular Probes, Eugene, OR, USA). The animals were anesthetized using isoflurane, and 10 μL 5% DiI solution in methanol were injected into the ear at several spots. Seven days later, the mice were sacrificed by CO_2_ inhalation, and TG were removed and examined fresh with a fluorescence microscope (Axioskop FS2, Zeiss, Jena, Germany).

### 2.3. Measurement of Scratching Behavior

For TNCB administration, we used a modified version of the protocol by Yamaura et al. [24], as illustrated in Figure 1. Briefly, on Day 9, the abdomens of the mice were shaved, and two days later, 100 μL 1.0% TNCB solution in acetone (*w*/*v*) was applied to the shaved area for sensitization. On Day 0, 10 μL TNCB solution were applied on each ear, and this was repeated on Days 2, 4, 7, 9, and 11. Phosphate buffered saline (PBS, 0.1 M, pH 7.4) was applied on the same days as the control.

Scratching behavior was recorded for 1.5 h, one day after each TNCB application. Videos were played back, and the number of scratching bouts was counted. A scratch bout is defined as lifting a hind paw directed to the ear or the head. This activity is distinct from stroking, which is typical for pain-associated behavior. To examine the effects of gap junction blockade on scratching behavior, the gap junction blockers carbenoxolone (CBX, 100 mg/kg, dissolved in saline) or meclofenamic acid (MFA, 10 mg/kg, dissolved in saline), both from Sigma-Aldrih (http://www.sigma-aldrich.com, accessed on 10 February 2022), were injected intraperitoneally (i.p.). Behavior was recorded 1.5 h after the injection.

### 2.4. Immunohistochemistry

The mice were killed with CO_2_ and both TG were removed and placed in 4% paraformaldehyde (PFA) in PBS for 1.5 h at room temperature (RT), followed by washing in PBS and incubation with 20% sucrose in PBS at 4 °C overnight. Then, ganglia were frozen in Tissue-Tek embedding medium and cut (10 μm thick) using a cryostat (Jung CM3000, Leica Microsystems, Wetzlar, Germany), mounted on slides, and washed with PBS prior to incubation with 50 mM NH_4_Cl for 0.5 h, at RT, to reduce autofluorescence. Sections were washed in PBS and incubated in blocking solution containing 3% bovine serum albumin (BSA) in PBS with 0.3% Triton X-100 for 2 h, at RT. The primary antibody against glial fibrillary acidic protein (rabbit anti-GFAP, Dako, Copenhagen, Denmark), was diluted 1:400 in 1% BSA, and incubated at 4 °C overnight. The primary antibody was replaced by 1% BSA in the control group. The sections were washed in PBS and incubated in secondary antibody (donkey anti-rabbit conjugated to Alexa Fluor 594 (Abcam, www.abcam.com, accessed on 10 February 2022) diluted 1:400 in PBS containing 1% BSA and 10 μM 4,6-diamidino-2-phenylindole dihydrochloride (DAPI, Sigma-Aldrich) to stain the nuclei for 2 h, at RT. Finally, slides were washed in PBS and visualized using an upright microscope (Axioskop FS2), equipped with fluorescent illumination and a digital camera (Penguin 600CL, Pixera, Los Gatos, CA, USA), connected to a personal computer. Microscope fields (315 × 235 μm) were selected randomly. All the images were acquired under identical conditions, and were analyzed in a blinded manner. Neuronal profiles containing the nuclei that were surrounded by GFAP-positive SGCs over 50% of their circumference were counted and expressed as a percentage of the total number of nuclei-containing neuronal profiles present in the field analyzed [25]. Four fields from the same ganglion were analyzed, and then averaged.

### 2.5. Dye Coupling

Freshly isolated ganglia were fixed with pins to the bottom of a silicon rubber-coated dish, and their capsule was removed. The dish was placed on the stage of an upright microscope, equipped with fluorescent illumination and a digital camera. The dish was superfused with Krebs solution, saturated with 95% O_2_ and 5% CO_2_, containing: 118 mM NaCl, 4.7 mM KCl, 14.4 mM NaHCO_3_, 1.2 mM MgSO_4_, 1.2 mM NaH_2_PO_4_, 2.5 mM CaCl_2_, and 11.5 mM glucose; pH 7.3. Individual SGCs and neurons in TG were injected with 3% fluorescent dye Lucifer yellow (LY, Sigma-Aldrich) in 0.5 M LiCl solution by sharp glass microelectrodes, which was connected to a preamplifier (Neuro Data Instrument Corp., New York, NY, USA). The dye was injected into the cells by hyperpolarizing current pulses, 100 ms in duration, 0.5 nA in amplitude, at 5 Hz for 3–5 min. Injections were made under visual inspection using a x40 water immersion objective to allow cell identification (neuron or glia) during the injection. The number of cells that were labeled as a result of dye passage from the injected cell (dye-coupled cells) were counted after the injection period. Coupling incidence was calculated as the ratio between the total number of injected cells to the number of dye-coupled ones. In some of the dye coupling experiments, we added the gap junction blockers: 50 μM CBX or 100 μM MFA, into the bathing solution 15 min prior to LY injection [26].

### 2.6. Ca^2+^ Imaging

Changes in the concentration of intracellular Ca^2+^ ([Ca^2+^]_in_) in SGCs were measured by microfluorimetry in intact ganglia. Freshly isolated ganglia were fixed to the bottom of a silicon rubber-coated dish (in Krebs solution), and their capsule was removed. Cells in the ganglia were loaded with the Ca^2+^ indicator Fluo-3 AM (10 µM, Invitrogen (www.invitrogen.com, accessed on 10 February 2022)) in minimum essential medium-α for 70 min in an incubator at 37 °C. Dishes were placed on the stage of an Axioskop FS microscope and superfused at 4 mL/min with Krebs solution (at 36–37 °C), saturated with 95% O_2_ and 5% CO_2_. ATP (5 μM) was applied by rapidly changing the bath solution. Images were acquired with a cooled CCD camera (PCO, Kelheim, Germany), using the Imaging Workbench 5 software (www.imagingworkbench.com, accessed on 10 February 2022). Fluorescence was excited at 450–490 nm, and emitted fluorescence (above 520 nm) was increased by elevated [Ca^2+^]_in_. Images were recorded at 0.3 Hz. The fluorescence ratio F/F_0_, where F_0_ is the baseline, was used to describe relative changes in [Ca^2+^]_in_.

### 2.7. Statistical Analysis

Dye coupling data were pooled for each time point from multiple experiments. This was done because in different dye coupling experiments, different numbers of cells were injected, and relatively small numbers of cells were injected per experiment. When an LY-injected cell was found to be dye coupled it was marked as 100, and when it was not coupled, it was marked as 0 [27]. These data were analyzed using one-way ANOVA with Tukey’s multiple comparison test. One-way ANOVA with Dunnett’s multiple comparison test was used to analyze the rest of the data. Values are expressed as mean ± SEM. *P* < 0.05 were deemed as statistically significant.

## 3. Results

### 3.1. Innervation of the External Ear by TG

As there is little information on TG afferents in the external ears in mice, we carried out retrograde labeling experiments to address this issue, using DiI injection into the ear. Seven days after the injections, we found clear labeling of neurons in the TG (Figure 2). This indicates that the external ear is innervated by TG neurons, but does not exclude additional pathways.

### 3.2. Itch Behavior in the Model Ear Atopic Dermatitis

The protocol for TNCB application is described in Figure 1. TNCB caused some reddening of the ear, but no lesions were observed. The assay for itch was the number of scratches per 1.5 h. In TNCB-treated mice, there was a significant increase in this parameter (Figure 3). We followed itch behavior after the last TNCB application (Day 11), and observed significant excessive scratching until Day 16, with a full return to baseline at Day 41 (Figure 3). The data for Figure 3 were obtained for females. To verify that the results were not sex dependent, we carried out a preliminary study on a group of males (*n* = 6) on Days 1–12, and the results were similar to those obtained for females (not shown).

### 3.3. SGC Activation

A hallmark of glial activation is GFAP upregulation. In numerous pain models, upregulation of GFAP expression in SGCs has been reported, for example, in [25]. We found that GFAP immunostaining in SGC increased significantly on Day 14, and decreased to control level on Day 41 (Figure 4).

### 3.4. Gap Junction-Mediated Cell Coupling

Increased gap junction-mediated coupling between SGCs has been observed in pain models, and appears to contribute to chronic pain [22,25,26,27,28,29,30]. We examined dye coupling between cells in TG from TNCB-treated mice. In control ganglia, 38.5% (15/39) of LY-injected SGCs were coupled to SGCs surrounding different neurons. On Days 10–12 TNCB applications, dye coupling incidence was 42.2% (19/45). There was no statistically significant difference between these groups (*p* > 0.05).

In contrast to the results on SGC coupling, there was a significant increase in neuron–neuron coupling on Days 10–12, which has been found in several pain models [26,28,30], see Figure 5. Adding gap junction blockers (CBX or MFA) to the bathing solution during the dye injections blocked neuron–neuron coupling (Figure 5C), supporting the idea that this coupling is mediated by gap junctions.

### 3.5. Gap Junctions Blockade and Itch Behavior

We asked whether blocking gap junctions would change the itch behavior. To test this, on Day 11 we injected the mice (i.p.) with CBX (50 mg/kg) or MFA (10 mg/kg), 1.5 h before the behavioral observations. We found that these drugs markedly reduced the number of scratches (Figure 6).

### 3.6. Calcium Imaging of SGC Responses to ATP

Under the current experimental conditions (Fluo-3 at 10 µM, intact ganglia), SGCs, but not neurons, are labeled by the calcium indicator. To label the neurons, a much higher concentration (920 µM) is required [31], and therefore, this method is selective for SGCs [32]. The sensitivity of SGCs to the P2 purinergic receptor agonist ATP has been found to be elevated in pain models [27,28]. We measured the responses of SGCs to ATP in intact TG from control mice and from TNCB-treated mice on Days 14, 18, 26 and 41. On Day 14, the response values were greater in ganglia from treated mice as compared with controls (Figure 7). The augmented responses persisted at day 18, and returned to baseline level on Day 41.

## 4. Discussion

Research in recent years has provided considerable evidence for the role of sensory ganglia in pain. The excitability of neurons in these ganglia is augmented in a variety of pain states and contributes to nociception [14,15,33]. The contribution of neurons and SGCs in sensory ganglia to chronic itch has received less attention. Andersen et al. [34] discussed the possibility that SGCs in sensory ganglia may participate in itch mechanisms, but direct evidence for this idea was not available. The present results show that both neurons and SGCs in TG underwent prominent changes in an itch model in mice and suggest that both cell types may play a role in pruritus.

Several types of animal models for itch have been described, some have been based on genetic engineering or inbred characteristics, and others have been based on applying substances to the skin [35,36]. Kitagaki et al. [23] developed a model for atopic dermatitis using repeated application of TNCB to the ears of mice. They found that BALB/C mice displayed atopic dermatitis-like phenotype, whereas C57BL/6 mice were less responsive. This model was simple and reproducible and was found to mimic many events occurring in the lesioned skin in human patients with chronic dermatitis. Other investigators have adopted this model (e.g., [24,37]). In the present work, we followed a TNCB protocol based on Yamaura et al. [24] with several modifications. Unlike Yamaura et al. [24] and others, we decided to use PBS as control rather than acetone. It has been reported that acetone does not induce scratching, but still has significant effects on the skin and its nerves [38]. There is evidence that even a minor skin lesion can affect SGCs [39], and because we were interested in identifying changes in the TG in an itch model, we preferred to avoid the damage induced by acetone. Due to the difference from previous studies, it is more accurate to describe our method as TNCB/acetone-induced itch model.

The human external ears are innervated, to a large extent, by nerve fibers that originate at the cervical spinal cord (great auricular nerve), but this topic is not entirely clear [40]. To test whether the TG innervates the ears in mice, we used retrograde labeling and found that neurons in the TG were labeled following a tracer injection in the ear. This validates the examination of the TG in the context of atopic dermatitis in the ear. The changes that we observed in this ganglion, which are discussed below, are consistent with this conclusion.

A hallmark of SGC activation is the upregulation of GFAP [25,41]. We found that, on Day 14 of the TNCB protocol, the number of TG neurons surrounded by GFAP-positive SGCs was greater by about 35% than that of controls. This increase was statistically significant, but smaller than that observed in a pain model based on lipopolysaccharide injection [28]. It should be mentioned that TNCB induces local inflammation [37], and therefore, the GFAP upregulation may not be specific for pruritus, but is a consequence of the inflammation [42]. For example, local inflammation of the sciatic nerve induced upregulation in GFAP immunostaining in SGCs in dorsal root ganglia [43]. Nevertheless, the results demonstrate that SGCs are altered in this itch model, and may contribute to the marked scratching behavior reported here (Figure 3). One possible mechanism by which activated SGCs can contribute to itch is to release substances that excite sensory neurons, such as ATP, or those that increase neuronal excitability, such as proinflammatory cytokines [20].

In all the pain models examined so far, a several-fold increase in gap junction-mediated coupling between SGCs was observed [25,26,27,28,29] and was correlated with upregulation of connexins (Cx), which are the gap junction proteins [44,45,46]. However, this topic needs to be further clarified, as there is evidence for the downregulation of Cx36 in dorsal root ganglia after sciatic nerve injury [47]. Coupling between neurons was also reported in pain models, but its magnitude was smaller than that of SGCs [26,28]. Here, we found that, unlike in pain models, in the itch model there was no change in SGC–SGC coupling; however, neuron–neuron coupling increased from about 2% to 13% (Figure 5). To test whether this coupling was mediated by gap junctions, we performed the dye coupling experiments while the bathing medium contained the gap junction blockers, CBX and MFA, and each separately blocked the augmented dye coupling, confirming the role of gap junctions in this effect. Thus, it may be proposed that the elevated neuronal coupling plays a role in pruritus.

To explore the possible role of cell coupling in scratching behavior, we injected TNCB-treated mice with CBX or MFA (i.p.) before measuring scratching behavior. We found that these drugs reduced scratching behavior, supporting the role of neuronal coupling in itch. The neuron–neuron coupling may lead to synchronous neuronal activity, thereby augmenting firing. A study on DRG neurons in vivo has shown that neighboring neurons fire synchronously (“coupled activation”) in mouse pain models and that gap junctions in SGCs contribute to this coupling and to the pain behavior in these models [48]. Coupled activation was explained by the possibility that neuronal coupling is mediated by gap junctions between SGCs [48], which was not observed in the present work. In contrast, in the itch model, neuron–neuron coupling appears to be direct, and it can be speculated that synchronous firing would be present without SGC coupling.

We have shown in a number of inflammatory pain models that the sensitivity of SGC to ATP is increased as compared with controls [27,28,49]. It was reported that ATP, acting on P2 purinergic receptors in MrgprA3-positive nerves, is involved in scratching behavior [50], suggesting a role for ATP in pruritus. We asked whether the sensitivity of SGC to ATP is changed in TNCB-treated mice. The results showed an increase in the response of these cells to ATP (Figure 7), indicating a similarity between changes in SGCs in pain models and in the itch model used here. This conclusion is supported by a study on diabetes-related pruritus in mice, which showed that the activation P2Y12 purinergic receptors induce SGCs activation, which, in turn, leads to the augmented scratching behavior of these mice [51].

If ATP plays a role in itch, we should ask what is the source of this substance. There is direct evidence that neurons in sensory ganglia release ATP upon firing, and ATP, in turn, acts on P2 receptors in SGCs [21,31,52]. Moreover, SGC themselves can release ATP, which would act on them in an autocrine manner and also on the neurons [21,52,53]. The greater sensitivity of SGCs to ATP described here, would augment this effect, leading to neuronal activation. SGCs also release cytokines, such as TNF-α, which can contribute to neuronal excitation [21,54]. It can be proposed that mediators released at the site of TNBS application can induce firing in the axons of sensory neurons, which would initiate the events described above, leading to enhanced activity of itch pathways.

The question of how pain and itch are encoded in sensory ganglia has not been resolved. One theory is the “labeled line” which proposes that itch- and pain-specific sensory neurons respond exclusively to their respective stimuli, each constituting a dedicated pathway [10]. However, alternative explanations, such as intensity and neuropeptide coding, have also gained some support [10,11,12]. A recent study suggested that coding of itch signals depended on the firing pattern of primary sensory neurons, which indicated that early processing in sensory ganglia was a major factor in pruritus [55]. The account above concerned acute itch and focused on the sensory neurons. To explain chronic itch, a role for spinal glia has been proposed [1,18]. It appears that astrocytes undergo prolonged activation in itch models [19], which may underlie the clinical picture. The present results show that both neurons and SGCs display distinct changes in the itch model, and are consistent with the idea that cellular interactions in sensory ganglia may play a role in both acute and chronic itch. According to the present results, some of the changes in neurons and SGCs persist for several days after the cessation of TNCB applications, which may correlate with the persistence of increase in scratching behavior that was observed.

## 5. Conclusions

TNCB induced several changes in TG: SGC activation, increased neuron–neuron coupling, and increased SGC response to ATP. The results indicate differences between changes in SGCs in pain and itch models; in the pain models, SGC coupling increased, whereas this did not occur in the itch model. However, in both models, neuron–neuron coupling increased, and both pain and itch behaviors were reduced by gap junction blockers. Thus, gap junction-mediated coupling apparently contributes to itch.

## Figures and Tables

**Figure 1 cells-11-00886-f001:**
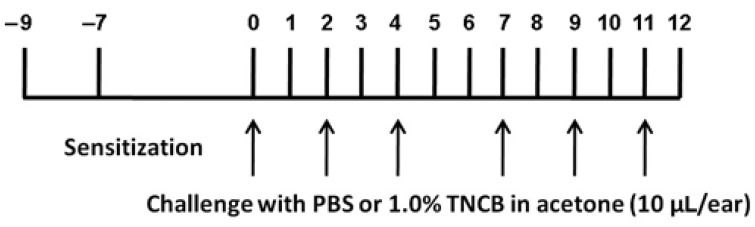
The experimental protocol for the induction of atopic dermatitis in mouse ears. TNCB was applied on both ears on the days indicated by the arrows. See text for further details.

**Figure 2 cells-11-00886-f002:**
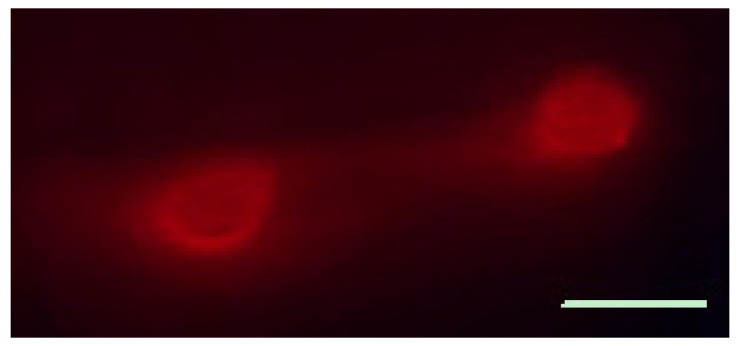
Trigeminal ganglion neurons innervate the external ear. The ganglion was examined one week after the fluorescent dye, i.e., 4,6-diamidino-2-phenylindole dihydrochloride (DiI), was injected into the ear. DiI-labeled neurons can be seen in the ganglion. Scale bar, 30 µm.

**Figure 3 cells-11-00886-f003:**
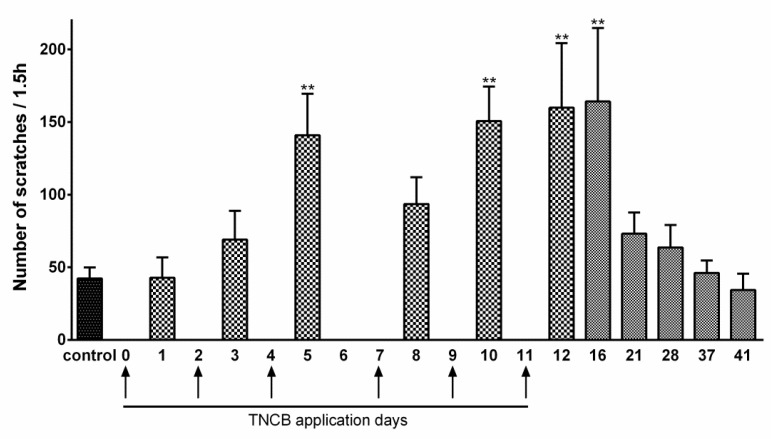
TNCB enhances scratching behavior. The application of TNCB was made from Day 0 to Day 11, as indicated by the arrows. The TNCB group is compared with the control group (saline). Saline application had no effect on scratching behavior from Day 0 to 12. The control value is the average of the measurements in saline-treated mice obtained on Days 1, 3, 5, 8, 10, and 12. The ordinate describes the number of scratches measured during 1.5 h one day after each TNCB application. *n* = 6 females. One-way ANOVA with Dunnett’s multiple comparison test, ** *p* < 0.01. Values are expressed as mean ± SEM.

**Figure 4 cells-11-00886-f004:**
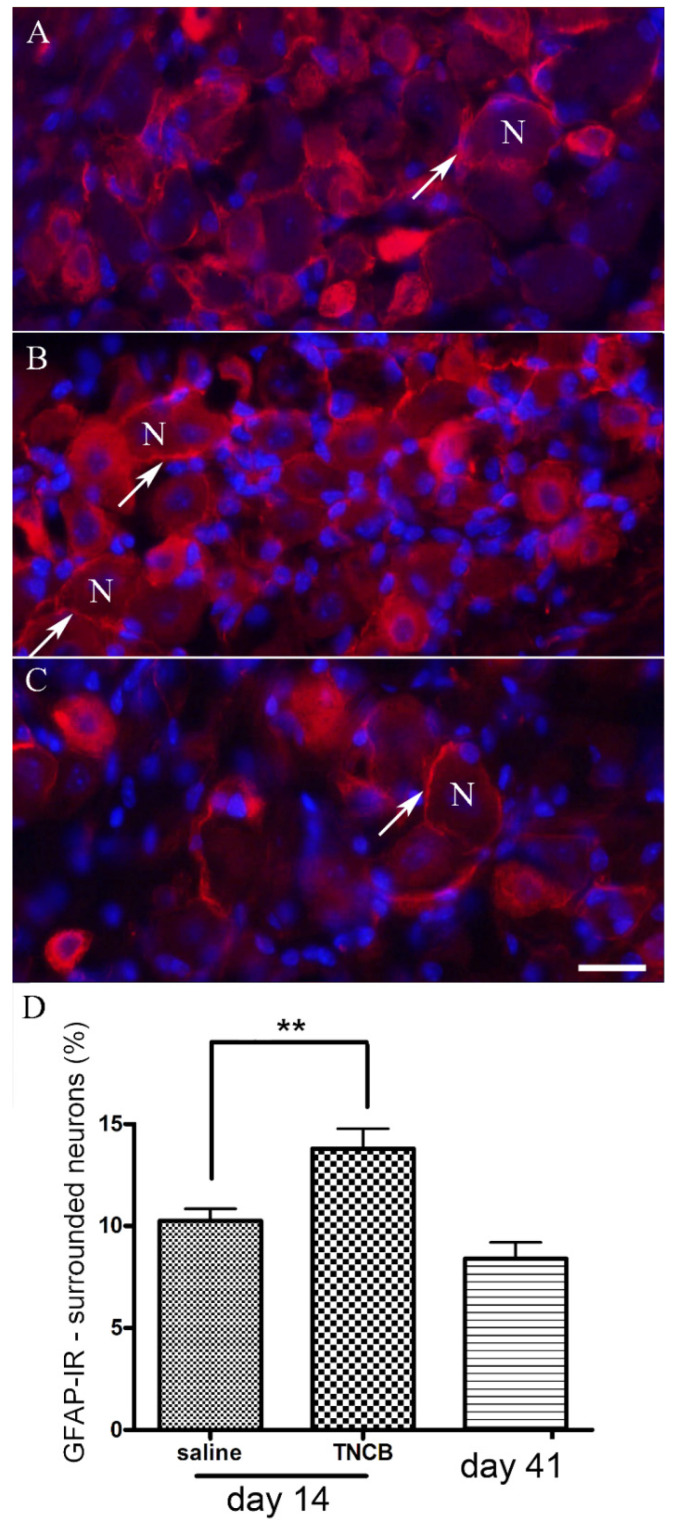
TNCB increases GFAP immunostaining in SGCs (shown in red). (**A**) An example of a section of TG in a control mouse (saline treatment); (**B**) section of TG from a mouse on Day 14 of the TNCB application protocol; (**C**) section of TG from a mouse on Day 41 (30 days after the last TNCB application). Cell nuclei are labeled blue with 4,6-diamidino-2-phenylindole dihydrochloride (DAPI). SGC nuclei are small and brightly labeled with DAPI, whereas neuronal nuclei are larger and weakly labeled. Several glial profiles stained for GFAP are indicated with arrows. The surrounded neurons are labeled with N. Calibration bar, 20 µm; (**D**) summary of results. *n* = 18 for the saline and TNCB groups, *n* = 10 for the post-TNCB group. ** *p* < 0.01, one-way ANOVA with Dunnett’s multiple comparison test. Values are expressed as mean ± SEM.

**Figure 5 cells-11-00886-f005:**
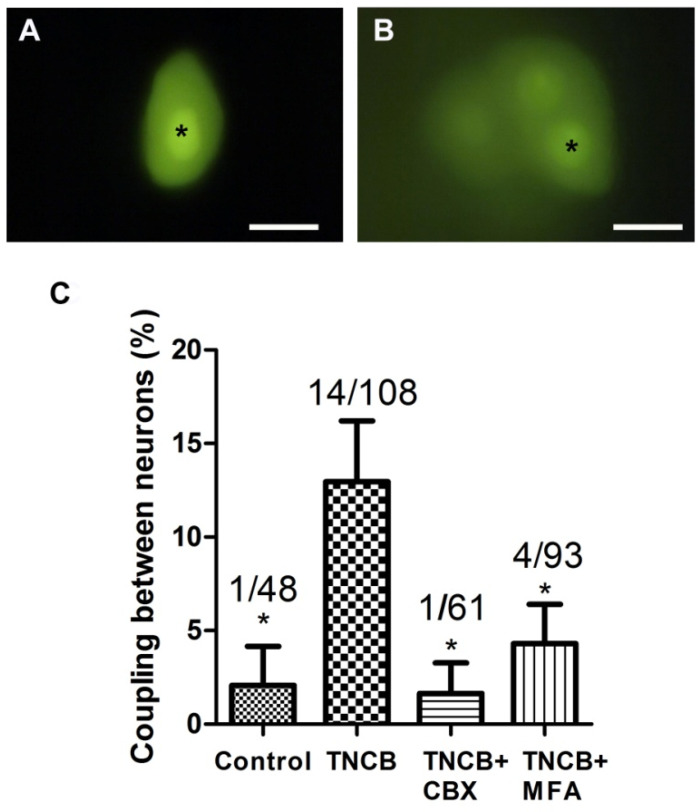
Dye coupling between neurons increases in TG from TNCB-treated mice. (**A**) Control, an LY-injected neuron is not coupled to other neurons; (**B**) dye coupling between neurons, observed in a TNCB-treated mouse; (**C**) summary of the incidence of neuron–neuron coupling and the effect of gap junction blockers on the augmented coupling after TNCB application. * Indicates the LY-injected neurons. Scale bars, 20 μm. The gap junction blockers were: carbenoxolone ((CBX) 50 μM) and meclofenamic acid ((MFA), 100 μM). One-way ANOVA with Tukey’s multiple comparison test,* *p* < 0.05. Values are expressed as mean ± SEM.

**Figure 6 cells-11-00886-f006:**
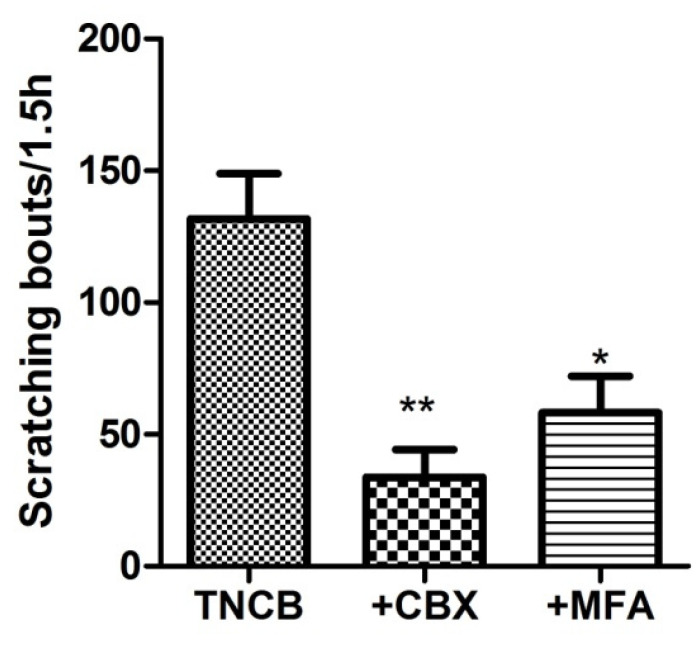
Gap junction blockers reduce scratching behavior in TNCB-treated mice. The gap junction blockers carbenoxolone (CBX, 50 mg/kg), or meclofenamic acid (MFA, 10 mg/kg), were injected intraperitoneally on Day 10, and scratching behavior was recorded 1.5 h afterward. CBX and MFA decreased the number of scratching bouts in treated animals; both comparisons are with respect to the TNCB-treated group. One-way ANOVA with Dunnett’s multiple comparison test, ** *p* < 0.01, * *p* < 0.05, *n* = 13 for TNCB group, *n* = 7 for TNCB +CBX group, and *n* = 6 for TNCB + MFA group. Values were expressed as mean ± SEM.

**Figure 7 cells-11-00886-f007:**
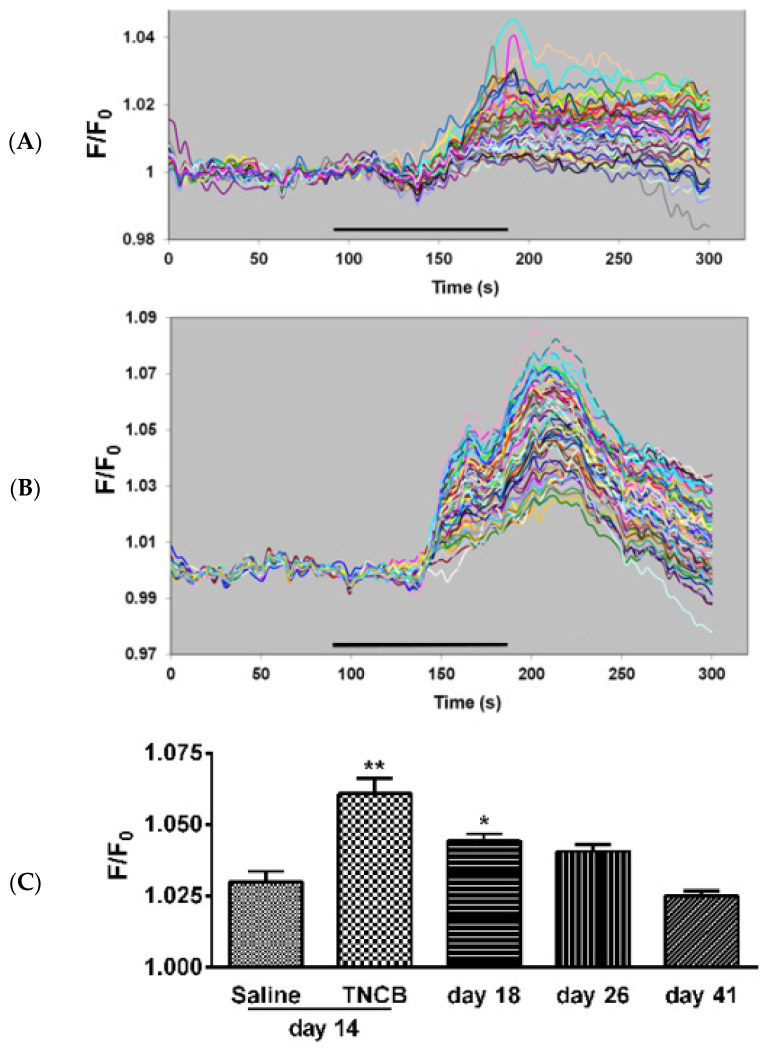
Ca^2+^ imaging of responses to ATP in SGCs. (**A**) Control, sample responses of SGCs to ATP (5 µM). Each trace is a response from a single cell; (**B**) responses from an animal on Day 14 of the TNCB application protocol; (**C**) summary of Ca^2+^ imaging experiments. The histograms compare responses to ATP (5 µM) of SGCs in TG from control mice (saline) and TNCB-treated mice. Ganglia from treated mice were examined until Day 41 (30 days after the last TNCB application). *n* = 4 females for each bar. The number of cells per bar was 185–240. Responses to ATP in ganglia from treated mice were greater than the control on Days 14 and 18 and returned to control value on Day 41. ** *p* < 0.01, * *p* < 0.05 as compared with saline. One way ANOVA with Dunnett’s multiple comparison test. Values are expressed as mean ± SEM.

## Data Availability

Data are available on request.

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
