# Peer review of "Satellite Glial Cells and Neurons in Trigeminal Ganglia Are Altered in an Itch Model in Mice"

_cells, 2022, doi:10.3390/cells11050886_

Round 1

Reviewer 1 Report

This is an interesting and original article which describes changes in trigeminal ganglia extracted from itchy mice compared to control mice.

This article is well organized, well written and referenced. The provided table and figures are informative.

There are a few errors and missing information that should be corrected.

P1, line 43: “pain and itch activate similar pathways” is confusing. This sentence should be reformulated.  Pain and itch pathways are not identical

P2, line 47: Another reference should be added to illustrate theories (A neuropeptide code for itch, Chen, Nature Reviews 2021)

P2, line 57: Replace “around the neurons” with “around the neuronal cell bodies”.

P6, figure 4: The GFAP labelling in A (control) appears higher than those in C (TNCB day 41). This is intriguing.

P9, lines 261-265: Font sizes have to be made uniform.

Author Response

P1, line 43: “pain and itch activate similar pathways” is confusing. This sentence should be reformulated.  Pain and itch pathways are not identical

Response

Thank you for your favorable words and the useful comments.

We wrote “similar”, not “identical”, but we agree that this statement  should be rewritten. We now mention that there is some overlap between pain and itch pathways, and that many pain- sensing sensory neurons can also transmit itch (Chen, Nature Reviews Neuroscience 2021).

2, line 47: Another reference should be added to illustrate theories (A neuropeptide code for itch, Chen, Nature Reviews 2021)

Response

This reference was added.

P2, line 57: Replace “around the neurons” with “around the neuronal cell bodies”.

Response

Corrected

P6, figure 4: The GFAP labelling in A (control) appears higher than those in C (TNCB day 41). This is intriguing.

Response

The image in 4C in the previous version was used because it  represented the number of GFAP positive SGCs at day 41 in a typical section, which is not significantly different from control (see the histogram). We agree that it was not representative of the background staining.  We selected another image  for Fig. 4C from the same experiment, which is representative of both GFAP positivity and background labeling.

P9, lines 261-265: Font sizes have to be made uniform.

Response

The font size is the same. This is a figure legend, and the line space is 1 instead of 1.5 in the text, to help the distinction between the two.

Reviewer 2 Report

Cohen et al. studied changes in neurons and satellite glial cells (SGC) in the trigeminal ganglia (TG) during chronic itch following chronic exposure of TNCB to ears of mice. They applied vehicle (PBS) or 1% TNCB in acetone solution to the ear of mice every other day for 12 days and observed and counted spontaneous scratching bouts for a 1.5 hr 24 hr following applications. They studied TGs from control and TNCB applied mice and examined glial activation and gap junctions. Based on their results they pretreated mice with two different gap junctions to understand whether those compounds would inhibit scratching behavior. They found that TNCB induces scratching and pretreatment with gap junction blockers reduces scratching significantly. They also found that an increase activation of GFAP immunoreactivity suggesting glial activation in mice with chronic itch. Additionally, authors showed an increase in neuron-neuron coupling but not SGC coupling in mice with scratching. The last, they investigated involvement of ATP in chronic itch and found that cells from itchy mice were more responsive than that of control mice. Even though results look novel, there are some questions need to be answered.

  1. Authors used PBS as vehicle in their studies even though TNCB was prepared in 1% concentration in acetone. Why did they not use PBS 1% in acetone?
  2. In animal section they are saying both female and male mice were used in the studies, however they are only showing female mice results? It would be better to use mixed gender or to show male mice data as well instead of saying there was no difference.
  3. 6 results for inhibition of scratching by both gap junction blockers, it was not mentioned that what the vehicle is for both compounds. Also, there should be another group showing vehicle-TNCB group.
  4. It would be better to show results for pain and itch in the same laboratory instead of referring results from other laboratories. They could have used cheek model of itch and pain to study and compare (described by Shimada SG, LaMotte RH 2008). Acute and following chronic applications chronic itch and pain could be studied.
  5. Introduction and discussion could have been written better.
  6. To show ATP as a mediator in this model to induce itch, P2Y12 receptor antagonist could have been studied both in vivo and in vitro for Calcium responses.

Author Response

  1. Authors used PBS as vehicle in their studies even though TNCB was prepared in 1% concentration in acetone. Why did they not use PBS 1% in acetone?

Response

Thank you for your helpful comments.

We did not use acetone as vehicle because acetone itself causes skin dryness, which can induce itch.  Therefore, acetone-treated mice would not be naïve and would not be proper controls.

  1. In animal section they are saying both female and male mice were used in the studies, however they are only showing female mice results? It would be better to use mixed gender or to show male mice data as well instead of saying there was no difference.

Response

We carried out these experiments on females because we followed the protocol of Yamaura et a (2012), who used females. The experiments on males were partial, and were done only to confirm that the results were not sex-dependent. The males were  followed only to day 12 and were therefore not included the full graph (to day 41). We added a statement explaining this point.

  1. 6 results for inhibition of scratching by both gap junction blockers, it was not mentioned that what the vehicle is for both compounds. Also, there should be another group showing vehicle-TNCB group.

Response

The vehicle was saline, and this is now mentioned in the Methods.

  1. It would be better to show results for pain and itch in the same laboratory instead of referring results from other laboratories. They could have used cheek model of itch and pain to study and compare (described by Shimada SG, LaMotte RH 2008). Acute and following chronic applications chronic itch and pain could be studied.

Response

We compared the itch results with results on pain models obtained  in  our laboratory (Hanani et al., 2002; Huang et al., 2010; Kushnir et al., 2011; Blum et al., 2014;  Warwick and Hanani, 2013;  Dublin and Hanani, 2007; Feldman-Goriachnik et al., 2015).

  1. Introduction and discussion could have been written better.

Response

We have improved the Introduction and Discussion following the comments of Reviewers 1 and  3.

  1. To show ATP as a mediator in this model to induce itch, P2Y12 receptor antagonist could have been studied both in vivo and in vitro for Calcium responses.

Response

As mentioned in the Introduction, this was designed as an initial work, which should be followed up. We agree that this is a good idea,  which will be tested in future work.

Reviewer 3 Report

The authors investigated the changes in neurons and satellite glial cells in the trigeminal ganglia in the TNCB induced itch mouse model. They found an enhanced itching in the TNCB model which could be blocked gap junction blockers. They showed that there was increased neuronal coupling (but not SBG cell coupling – data not shown) in the TNCB treated mice. They also found increased calcium influx in response to ATP in SGC from TNCB treated animals compared to controls.

A few comments are below:

Figure 2.  I would suggest a better representative image be shown, including neuronal markers or smaller magnification which shows the percentage of DiI labeled neurons in the TG.  I think this is important for readers to get an idea of the % of TG neurons innervating the ear.

Figure 4 shows a dramatic reduction in GFAP-IR at day 41 which appears significantly different from control.  Was this assessed for statistical significance? Is this a representative image and is this reduction? I would have liked to see GFAP co-stained with neuronal-specific markers to label neurons

What percentage of SCG coupling was observed in both cases (control and TNCB treated) in the TG. Data was not shown but mentioned a number of times.

They saw enhanced SGC activity upon stimulation with ATP, yet you saw no evidence of coupling, what is your explanation or proposed mechanism for enhanced SGC activity.  How were you certain that the calcium imaging was of SGC and not neurons?

Did you check for upregulation of connexins?

Sentence lines 304-307 “The neuron-neuron coupling may lead to synchronous neuronal activity, thereby augmenting firing. A study on DRG neurons in vivo has shown that neighboring neurons fire synchronously ("coupled activation") in mouse pain models and that gap junctions in SGCs contribute to this coupling and to the pain behavior in these models [43].” Seems a bit contradictory as you already mentioned that you saw no change in SGC coupling.  If you are suggesting the SGC has other effects on neurons unrelated to SGC coupling, I would suggest a more detailed discussion here on the proposed mechanism.

The discussion reads like a repeat of the results section.  I would suggest that authors discuss, the possible sources of ATP and how and why they think the SGCs have an increased response to ATP. 

Did they also observe whether there was increased neuronal activity in TCNB treated animals and was this blocked by gap junction blockers?

Author Response

Figure 2.  I would suggest a better representative image be shown, including neuronal markers or smaller magnification which shows the percentage of DiI labeled neurons in the TG.  I think this is important for readers to get an idea of the % of TG neurons innervating the ear.

Response

The retrograde labeling was done solely to verify that the external ears of mice receive innervation from the trigeminal ganglion. What the reviewer is suggesting is indeed interesting, but is not relevant to the main message of this work, which is to shed light on some functional and chemical changes in neurons and SGCs in an itch model.

Figure 4 shows a dramatic reduction in GFAP-IR at day 41 which appears significantly different from control.  Was this assessed for statistical significance? Is this a representative image and is this reduction? I would have liked to see GFAP co-stained with neuronal-specific markers to label neurons

Response

The image in 4C in the previous version was used because it  represented the number of GFAP positive SGCs at day 41 in a typical section, which is not significantly different from control (see the histogram). We agree that it was not representative of the background staining. We selected another image from the same experiment, which is representative of both GFAP positivity and background labeling.

We now included nuclear staining with DAPI to show that neurons and SGCs could be distinguished. SGC nuclei are small and brightly stained with DAPI. Neuronal nuclei are larger and much paler. This is now mentioned in the Figure 4 legend.

What percentage of SCG coupling was observed in both cases (control and TNCB treated) in the TG. Data was not shown but mentioned a number of times.

Response

We examined dye coupling in TG from TNCB-treated mice. In control ganglia, 38.5% (15/39) of LY-injected SGCs were coupled to SGCs surrounding different neurons. At 10–12 days of TNCB applications, dye coupling incidence was 42.2% (19/45). There was no statistically significant difference between these groups (p>0.05).  This is included now in the Results.

They saw enhanced SGC activity upon stimulation with ATP, yet you saw no evidence of coupling, what is your explanation or proposed mechanism for enhanced SGC activity.  How were you certain that the calcium imaging was of SGC and not neurons?

Response

The most likely explanation for the enhanced response of SGC to ATP is the upregulation in P2 receptors in these cells. This topic has been investigated by us in detail previously in inflammatory pain models (Kushnir 2011; Feldman 2015). This issue is now discussed and the relevant references have been added.

Under the current experimental conditions (Fluo-3 at 10 µM, intact ganglia) SGCs, but not neurons are labeled by the calcium indicator. To label the neurons a much higher concentration (920 µM) is required (Zhang et al., 2007), and therefore this method is selective for SGCs. This topic has been discussed by us in the first paper on Ca imaging in intact sensory ganglia (Weick et al., 2003).   This is now mentioned in the section on Ca imaging.

Did you check for upregulation of connexins?

Response

We did not check changes in connexins. This is a large topic that should be the subject of future work. As we stated in the Introduction, this work was designed to be an initial study on changes in a sensory ganglion in an itch model. Indeed, this paper opens up questions that will be addressed in future work, and we hope that other groups will contribute to this effort.

Sentence lines 304-307 “The neuron-neuron coupling may lead to synchronous neuronal activity, thereby augmenting firing. A study on DRG neurons in vivo has shown that neighboring neurons fire synchronously ("coupled activation") in mouse pain models and that gap junctions in SGCs contribute to this coupling and to the pain behavior in these models [43].” Seems a bit contradictory as you already mentioned that you saw no change in SGC coupling.  If you are suggesting the SGC has other effects on neurons unrelated to SGC coupling, I would suggest a more detailed discussion here on the proposed mechanism.

Response

Coupled activation was described as a synchronous activity of neighboring sensory neurons (Kim et al., 2016). The mechanism underlying coupled activation is not entirely clear, but it seems that SGC gap junctions are essential for it. The explanation that we offered for the role of SGC gap junctions in coupled activation is that SGCs serve as a bridge between neurons, enabling neuronal electrical coupling (Kim et  al., 2016). Thus, the absence of SGC coupling would appear to contradict the explanation offered in the present paper. However, we presented  here  clear neuron-neuron coupling due to TNCB application, which is in agreement with the explanation of coupled activation, except that augmented SGC coupling apparently does not make an important  contribution in the case of itch. The direct neuron-neuron coupling observed here seems a more likely option. This point is now included in the discussion.

As for the role of SGC in itch, the observations that they are activated (GFAP upregulation) and their response to ATP is elevated, indicate that SGC activation may contribute to itch. We now discuss this point.

The discussion reads like a repeat of the results section.  I would suggest that authors discuss, the possible sources of ATP and how and why they think the SGCs have an increased response to ATP. 

Response

The discussion includes the topic of ATP and additional points mentioned above.

Did they also observe whether there was increased neuronal activity in TCNB treated animals and was this blocked by gap junction blockers?

Response

We did not study this topic.

Round 2

Reviewer 2 Report

Dear Authors,

Thank you for your responses to questions and comments. Your answer to my question related to use of acetone as vehicle is not satisfying. Acetone as vehicle has been used in dermatitis studies for TNCB in previous reports like Matsukura et al. (Int Arch Allergy Immunol 2005;136:173-180), Harada et al. (J of Dermatological Science 2005;37, 159-167) and even the one you are saying that you followed the protocol Yamaura et al. (2012, ref 24).  Chronic topical exposure of acetone to skin of mouse either on ear or behind the neck does nor induce any dry and itchy skin. Since this is the first study examining changes in TN and satellite glial cells during the AD, it is very crucial that the observed effects are not from the vehicle. Using acetone as vehicle, TNCB application related immunohistochemistry studies needs to be conducted.

Author Response

Thank you for your comment.

We fully understand and respect your views on this matter. We were aware that acetone has been used as control for the TNCB method, but preferred not to do that. We believe that in certain cases using the vehicle as control may present disadvantages.  There is evidence that acetone induces significant changes in the nerves in the skin (see below*). Our main interest in this work was to examine changes in TG neurons and particularly, satellite glial cells (SGCs), There is evidence that SGCs are very sensitive to even minor skin lesion [see e.g., Glia 45 (2004) 105– 109.] We were therefore concerned that acetone application might mask changes associated with itch. We now address this point in the Discussion. We also clarify that our model is based on the combined effect of TNCB/acetone.  

*Moniaga et al., Mechanisms and Management of Itch in Dry Skin. Acta Derm Venereol. 2020;100(2):adv00024. doi: 10.2340/00015555-3344.

“Others observed the release of histamine from mast cells in the skin of acetone-treated mice (25). We found that acetone-treated mice displayed a rapid increase in TEWL [transepidermal water loss] and a decrease in SC hydration during the first hour after treatment, which returned to normal by 48 h after the treatment. Thus, the acetone-treated mice manifest the characteristics of dry skin and have altered cutaneous barrier permeability. No scratching behaviours or epidermal hyperplasia were observed in the acetone-treated mice, although there was an increase in nerve fibre density in the epidermis.”

Reviewer 3 Report

Since the title of the article is "Neurons and satellite glial cells in trigeminal ganglia are altered in an itch model in mice",  I would have liked to see you address my last question as to whether the activity/sensitivity of the neurons themselves was altered in this itch model as you did with the calcium imaging of the satellite glial cells, and whether this was dependent on gap-junctions.

Author Response

Thank you for your comment. We agree that this point is of interest, but it is beyond the scope of this work. What you suggest is a project on its own. In principle, calcium imaging of TG neurons would be a good way to go, but neurons in intact ganglia are not labeled with calcium indicators. Doing electrical recording is a major undertaking that would require many months, but we were given only 10 days by the Editor to complete the revision.

The previous title of the paper started with the neurons. We made a small  change in the title to put more emphasis on the glia:  “Satellite glial cells and neurons in trigeminal ganglia are altered in an itch model in mice”.